# Omega-3 Polyunsaturated Fatty Acids (PUFAs) and Diabetic Peripheral Neuropathy: A Pre-Clinical Study Examining the Effect of Omega-3 PUFAs from Fish Oil, Krill Oil, Algae or Pharmaceutical-Derived Ethyl Esters Using Type 2 Diabetic Rats

**DOI:** 10.3390/biomedicines13071607

**Published:** 2025-06-30

**Authors:** Eric Davidson, Oleksandr Obrosov, Lawrence Coppey, Mark Yorek

**Affiliations:** 1Department of Internal Medicine, University of Iowa, Iowa City, IA 52242, USA; eric-davidson@uiowa.edu (E.D.); oleksandr-obrosov@uiowa.edu (O.O.); lawrence-coppey@uiowa.edu (L.C.); 2Department of Veterans Affairs Iowa City Health Care System, Iowa City, IA 52246, USA; 3Veterans Affairs Center for the Prevention and Treatment of Visual Loss, Iowa City, IA 52246, USA; 4Fraternal Order of Eagles Diabetes Research Center, University of Iowa, Iowa City, IA 52242, USA

**Keywords:** omega-3 polyunsaturated fatty acids, fish oil, peripheral neuropathy, eicosapentaenoic acid, docosahexaenoic acid

## Abstract

**Background and Objectives**: We have previously reported that omega-3 polyunsaturated fatty acids (PUFAs) derived from fish oil (FO) is an effective treatment for type 1 and type 2 diabetes neural and vascular complications. As omega-3 PUFAs become more widely used as a nutritional and disease modifying supplement an important question to be addressed is what is the preferred source of omega-3 PUFAs? **Methods**: Using a type 2 diabetic rat model and early and late intervention protocols we examined the effect of dietary treatment with omega-3 PUFAs derived from menhaden (fish) oil (MO), krill oil (KO), algal oils consisting primarily of eicosapentaenoic acid (EPA), docosahexaenoic acid (DHA) or combination of EPA + DHA, or pharmaceutical-derived ethyl esters of EPA, DHA or combination of EPA + DHA. Nerve related endpoints included motor and sensory nerve conduction velocity, heat sensitivity of the hind paw, intraepidermal nerve density, cornea nerve fiber length, and cornea sensitivity. Vascular reactivity to acetylcholine and calcitonin gene-related peptide by epineurial arterioles that provide blood to the sciatic nerve was also examined. **Results**: The dose of each omega-3 PUFA supplement increased the content of EPA, docosapentaenoic acid (DPA), and/or DHA in red blood cell membranes, serum and liver. Diabetes caused a significant decrease of 30–50% of neural function and fiber occupancy of the skin and cornea and vascular reactivity. Treatment with MO, KO or the combination of EPA + DHA provided through algal oil or ethyl esters provided significant improvement of each neural endpoint and vascular function. Algal oil or ethyl ester of EPA alone was the least effective with algal oil or ethyl ester of DHA alone providing benefit that approached combination therapies for some endpoints. **Conclusions**: We confirm that omega-3 PUFAs are an effective treatment for DPN and sources other than fish oil are similarly effective.

## 1. Introduction

The diagnosis of diabetic peripheral neuropathy (DPN) remains chiefly a clinical one with symptoms and signs reflective of the underlying damage to different types of nerve fibers [1]. The classic presentation often referred to as “stocking-and-glove” distribution begins at the tips of the toes with sensory nerve changes that include symptoms like numbness, tingling, burning, or pain and advances in a symmetrical, distal-to-proximal pattern [1]. In more advanced stages, large fiber damage can result in numb, insensate feet, predisposing the patient to the development of diabetic foot ulcers with high risk of infection and amputation, as well as reduced daily functioning such as poor balance leading to falls and fractures [1].

About 50% of diabetes patients both type 1 and type 2 will be affected by DPN [2,3]. Thus, there is a critical need of finding a safe and efficacious means of treatment. There is no FDA approved treatment for DPN. Due to the complexity of the etiology of DPN that includes increased inflammatory and oxidative stress and mitochondrial dysfunction a successful treatment paradigm will likely need to include a combination of lifestyle changes along with drug and dietary interventions [4]. Toward that end we have been examining the effect of long-chain omega-3 (n-3) polyunsaturated fatty acids (PUFAs), (primarily eicosapentaenoic acid (EPA) and docosahexaenoic acid (DHA)) that is enriched in fish oil and their metabolites as a treatment for microvascular and neural complications associated with type 1 and type 2 diabetes [5,6,7,8].

As research continues to unveil the importance of omega-3 PUFAs in healthy aging and management of diseases, issues such as sustainability, quality control, and environmental concerns raises the question whether fish oil is the only efficacious source [9]?

## 2. Material and Methods

Supplies for the experiments described were obtained from Sigma-Aldrich (St. Louis, MO, USA) unless stated otherwise.

### 2.1. Animals, Diets and Experimental Design

Male Sprague-Dawley (Envigo, Indianapolis, IN, USA) rats 10-11 weeks of age were housed in a AAALAC accredited animal housing facility and diets (Teklad, #7001, Madison, WI, USA (control diet)) and water were provided ad libitum. This study was approved by the Iowa City VA Healthcare system Institutional Animal Care and Use Committee (ACORP 2190501). Guide for the Care and Use of Laboratory Animals as well as VA Directive 1200.07 were followed. 

Diabetes was induced through a combination of a high fat diet (D12451; Research Diets, New Brunswick, NJ, USA) for 8 weeks followed by a low dose of streptozotocin (30 mg/kg) with an overnight fasting [5,7]. Control rats were treated with vehicle. Feeding the rats a high fat diet renders them insulin resistant and treating them with a low dose of streptozotocin destroys a portion of their β-cells making them unable to produce sufficient insulin to overcome the insulin resistance rendering them hyperglycemic [10,11]. Diabetes was confirmed 3 days post streptozotocin injection and rats with greater 250 mg/dl blood glucose were deemed diabetic. 

Dietary omega-3 PUFAs intervention treatments were initiated after 4 weeks (early protocol) or 10 weeks (late protocol) of hyperglycemia. These diets were enriched with menhaden (fish) oil (D16021504), krill oil ((Jedwards International, Braintree, MA, USA) (D21062302)), or 3 different algal oils derived from algae that produce primarily EPA, AzAlgae EPA15+ crude extract ((Arizona Algae Products, Holbrook, AZ, USA) (D21062303)), primarily DHA, Life’s DHA S40 – O400 (DSM Nutritional Products, Columbia, MD, USA) (D21062304)), or the combination of EPA + DHA, Life’s Omega 45 O2412 – O100 (DSM Nutritional Products, Columbia, MD, USA) (D21062305)). These diets were created by replacing ½ kcal of the 45% high fat diet with the respective oil (app. 1420 g oil per 12.5 kg diet). Dietary interventions with omega-3 PUFAs derived from ethyl esters were created by adding 210 g of MEG-3 ultra-high EPA EE oil (DSM Nutritional Products, Columbia, MD, USA) (D21062306), MEG-3 0070 EE oil (DSM Nutritional Products, Columbia, MD, USA) (D21062305), or MEG-3 5020 EE oil (DSM Nutritional Products, Columbia, MD, USA) (D21062307) to the 45% kcal high fat diet. This provided the ethyl ester-based diets for EPA, DHA and EPA + DHA, respectively. The appropriate amount of each of the ethyl ester oils was determined through a preliminary study. The target for the dietary interventions was to obtain a combined increase in EPA, DPA, and DHA in red blood cell membranes greater than 8%. In human subjects a “healthy” omega-3 index is defined as a range between 8–12% of EPA and DHA in red blood cell membranes [12]. In this pre-clinical study, we have included levels of DPA with EPA and DHA because of the changes in the levels of all 3 of these omega-3 fatty acids observed during dietary intervention. All diets were made by Research Diets (New Brunswick, NJ, USA) by providing them with the appropriate amount of each oil for the individual diets. Treatment period for both early and late intervention protocols was 12 weeks. The study of menhaden oil, krill oil, and the three algal oil omega-3 PUFAs were combined in one study group and based on a power analysis repeated a minimum of three times with three rats for each condition. Likewise, we combined menhaden oil with the three-ethyl ester omega-3 PUFAs in a separate study and based upon a power analysis repeated a minimum of three times with three rats for each condition. Treatment with menhaden oil was included in both study groups as a positive control. Appendix A provide data for the fatty acid composition of these diets.

### 2.2. Endpoints Related to Nerve and Vascular Reactivity

The methods used for the neural and vascular endpoints reported in this study were procedures common to the laboratory [5,7,8]. These endpoints included sensitivity to heat applied to the hindpaw, motor and sensory nerve conduction velocity, corneal nerve fiber length, cornea nerve sensitivity, intraepidermal nerve fiber density and vascular reactivity to acetylcholine and calcitonin gene related peptide (CGRP) of epineurial arterioles that provide blood flow to the sciatic nerve.

### 2.3. Fatty Acid Composition

Fatty acid composition of the diets, serum and liver were measured after the lipid fraction was extracted using chloroform/methanol, followed by transesterification in 14% boron trifluoride in methanol and extraction of the fatty acid methyl esters into heptane. The fatty acids were then separated by gas-liquid chromatography [13]. Individual fatty acids peaks as % of total fatty acids present were identified by comparison to known fatty acid standards. Extraction of lipids from red blood cells was performed following the method of Siscovick et al. [14].

### 2.4. Data Analysis 

Data are presented as mean ± SEM. Statistical analysis, one-way ANOVA and Bonferroni posttest comparison, was performed using GraphPad Prism software version 10.5.0 (San Diego, CA, USA). Responsiveness to acetylcholine and CGRP were compared applying a two-way repeated measures analysis of variance with autoregressive covariance structure using Sigma Plot version 16.0 (Palo Alto, CA, USA) [5,7,8]. A *p* value of less than 0.05 was deemed significant.

## 3. Results

### 3.1. Status of Animal Weight, Blood Glucose, Omega-3 Index and Fatty Acid Composition in Serum and Liver

At the beginning of the study rats were 12 weeks of age and weighed an average of 325 ± 2 grams. At the end of the study period control rats in the early and late intervention groups gained about 165 and 205 grams, respectively. Diabetic rats also gained weight over the study period, but their end weight was significantly less than control rats. All diabetic rats were hyperglycemic throughout the study and significantly increased vs. control rats at the end of the study. Treatment with omega-3 PUFAs did not reduce elevated glucose levels in diabetic rats. Serum free fatty acids and cholesterol levels were significantly increased in diabetic rats included in the early intervention protocol and serum free fatty acids, triglycerides and cholesterol in the late intervention protocol. Treating diabetic rats with menhaden oil, krill oil or EPA alone derived from either algal oil or as an ethyl ester did not reduce serum cholesterol. There was a trend to reduce serum cholesterol with treatment of DHA alone from algal oil or EPA + DHA derived from either algal oil or as an ethyl ester. Serum triglyceride levels were significantly decreased in diabetic rats that were treated with DHA alone and EPA + DHA derived from algal oil in the late protocol. Serum free fatty acids were significantly increased in diabetic rats and generally not significantly improved compared to untreated diabetic rats with treatment by omega 3 PUFAs except for treatment with DHA alone derived from algal oil (Appendix A).

The fatty acid composition of red blood cell membranes and summation of EPA, DPA and DHA is provided in Table 1. These data were derived from rats after 12 weeks of dietary treatment. Levels of EPA + DPA + DHA for control and diabetic rats was 2.7% and 3.4%, respectively. The levels of omega-3 fatty acids were significantly increased by each of the sources of omega-3 PUFAs. EPA and DHA alone derived from algal oils or ethyl esters had a lesser impact than menhaden oil, krill oil, or EPA + DHA derived from algal oil or ethyl ester. These data demonstrate that dietary manipulation with different sources of omega-3 PUFAs can significantly increase the omega-3 fatty acid composition of red cell membranes [12].

Data of the fatty acid composition of serum and liver for control, untreated diabetic rats, and diabetic rats treated with the different sources of omega-3 PUFAs for 12 weeks are provided in Appendix A, respectively. The unsaturation index defined as the mean number of double bonds present was significantly increased in serum from diabetic rats dietarily treated with menhaden oil, krill oil, and each of the diets containing the three algal oils enriched in EPA, DHA or the combination of EPA + DHA, and EPA + DHA derived from ethyl ester compared to serum from untreated diabetic rats. The ethyl ester of EPA or DHA alone did not significantly change the unsaturation index of serum. Further analysis of the serum from diabetic rats treated with the different sources of omega-3 PUFAs revealed that only EPA or DHA were increased in serum when treatment was with algal oils or pharmaceutical derived ethyl esters containing only EPA or DHA, respectively. When diets were enriched with both EPA + DHA derived from menhaden oil, krill oil or algal oils or ethyl ester that contain both EPA + DHA the levels of EPA and DHA were both increased. DPA was significantly increased in serum when diabetic rats were treated with EPA alone derived from either algal oil or ethyl ester indicating elongation of EPA.

Analysis of the liver revealed that the unsaturation index was significantly increased with diets containing menhaden oil, krill oil or algal oils enriched with DHA or EPA + DHA compared to untreated diabetic rats. Treatment of diabetic rats with EPA alone derived from algal oil or with EPA, DHA, or EPA + DHA derived from ethyl ester did not significantly increase the unsaturation index because the increase of the omega-3 PUFAs in the liver was offset by a decrease in omega-6 PUFAs. The individual increases of EPA and DHA in the liver was like what was observed in the serum and corresponded to the EPA and DHA composition of the diet. 

### 3.2. Effect on Motor and Sensory Nerve Conduction Velocity, Thermal and Cornea Sensitivity, Intraepidermal and Corneal Nerve Fiber Density

Table 2 and Table 3 provide data of the effect of treating diabetic rats with diets enriched in menhaden oil, krill oil, and algal oils containing primarily EPA, DHA or EPA + DHA on DPN related endpoints following early and late intervention, respectively. Diabetes caused a significant impairment in each of the neuropathy related endpoints. Early and late intervention with menhaden oil, krill oil, and algal oils providing DHA alone or EPA + DHA significantly improved each of these endpoints compared to untreated diabetic rats. However, some of these treated endpoints remained significantly decreased compared to control rats. In contrast, treatment with algal oil of EPA alone was less beneficial providing significant improvement in heat sensitivity of the hindpaw, intraepidermal nerve fiber density, and cornea sensitivity but minimal improvement in motor and sensory nerve conduction velocity. 

Table 4 and Table 5 present data of the effect of early and late intervention treatment, respectively, of manufactured omega-3 PUFAs provided in the form of ethyl esters of EPA, DHA and EPA + DHA on DPN related endpoints. As a positive control treatment with menhaden oil was included. Motor and sensory nerve conduction velocities, heat sensitivity of the hindpaw, intraepidermal density, cornea nerve fiber length, and cornea sensitivity were significantly improved by ethyl esters of DHA alone and EPA + DHA in both the early and late intervention protocols. Treatment with ethyl ester of EPA alone was less effective in comparison. 

### 3.3. Effect on Vascular Reactivity to Acetylcholine and CGRP by Epineurial Arterioles Providing Blood Flow to the Sciatic Nerve 

Data for relaxation of epineurial arterioles by acetylcholine and calcitonin gene-related peptide (CGRP) are shown for the late intervention protocol. Similar results were observed in the early prevention protocol (data not reported). Diabetes causes a significant decrease in vascular relaxation to acetylcholine (Figure 1 and Figure 2). Treating diabetic rats with menhaden oil, krill oil or the algal oil of EPA + DHA significantly improved vascular relaxation (Figure 1). Treatment with algal oils of EPA or DHA alone were less effective and relaxation remained significantly impaired compared to control. Data in Figure 2 show that treating diabetic rats with ethyl ester of DHA alone or EPA + DHA significantly improved vascular relaxation to acetylcholine. In contrast, treating diabetic rats with ethyl ester of EPA alone provided minimal improvement. CGRP is a vigorous vasodilator in epineurial arterioles [15]. Vascular relaxation to CGRP was significantly decreased in diabetic rats compared to control rats (Figure 3 and Figure 4). Treatment with menhaden oil, krill oil and algal oils of DHA alone and EPA + DHA significantly improved vascular relaxation to CGRP while treatment with algal oil of EPA provided no improvement (Figure 3). Data in Figure 4 demonstrate that treating diabetic rats with ethyl ester of DHA and EPA + DHA significantly improved vascular relaxation to CGRP, and ethyl ester of EPA showed some improvement but remained significantly impaired compared to control rats. In this set of studies that were repeated 3 times it was observed that the effect of menhaden oil as the positive control was not as robust as observed in the other study group.

## 4. Discussion

The number of publications regarding omega-3 PUFAs increased 60% from 2010 to 2020 due to the expanding interest of their potential benefits in healthy aging and multiple diseases [16,17,18]. Long-term supplementation of fish oil capsules or occasionally krill oil is generally safe, with a few exceptions, and has been shown to potentially benefit cardiovascular disease, Alzheimer’s disease, cognitive dysfunction, type 2 diabetes, dyslipidemia, insulin resistance, muscle atrophy, cancer, and fetal development [16,17,18,19,20]. With the growing interest in omega-3 PUFAs as a nutritional supplement with potential health benefits an important question is whether capsulated fish oil is the best source of omega-3 PUFAs? The Food and Drug Administration regulates prescription ethyl ester omega-3 PUFAs supplements like Lovaza and Vascepa, but not fish oil capsules sold over-the counter. This raises questions about the quality of the products being sold regarding the marketed content of omega-3 PUFAs especially if achieving a healthy omega-3 index is the goal of the consumer. Many of the fish used to provide omega-3 PUFAs oils are farm raised and their environmental condition and diet can influence the quantity and quality of omega-3 PUFAs harvested. Considering these limitations and the potential of an expanding demand are their other sources of omega-3 PUFAs that are more environmentally friendly, sustainable, reliable, and safe?

The most common omega-3 PUFA is α-linolenic acid, which is present in some vegetable oils, nuts, flaxseeds, and soy products [21]. α-Linolenic acid can be modified by a series of elongation and desaturation reactions to form longer chain omega-3 PUFAs such as EPA and DHA. A healthy omega-3 index range is 8–12% but most people in the United States have an index that ranges from 3–4% [22]. Suggesting that the elongation and desaturation of α-linolenic acid is not sufficient to achieve a healthy omega-3 index for most people and can only be reached through a modified diet or supplementation. A recent study demonstrated that daily consumption of 30 g/day of ground flaxseed significantly improved several cardiometabolic risk factors, including body weight, body mass index, lipid levels, blood pressure, glycemic measures, markers of inflammation (e.g., C-reactive protein and interleukin-6), oxidative stress, and liver enzymes [23]. However, consumption of this amount of a daily supplement is likely not sustainable for most people. A more acceptable approach for increasing long chain omega-3 PUFAs is supplementation with direct sources of EPA and DHA that are present in fish oil, krill oil, algae, yeast, and pharmaceutically derived compounds [24]. In this study we sought to determine whether these different sources of omega-3 PUFAs could be used interchangeably as a treatment for DPN in a rat model of late-stage type 2 diabetes.

Treating diabetic rats with the different sources of omega-3 PUFAs caused significant increases in the omega-3 fatty acid levels in red blood cells, serum and liver, compared to control and non-treated diabetic rats that corresponded to the fatty acid composition of the diet. Treatment with algal oils or ethyl esters of EPA or DHA alone resulted in lower omega-3 fatty acid levels compared to treatment with menhaden oil, krill oil or algal oil or ethyl ester of EPA + DHA. Treatment with algal oil or ethyl ester of EPA alone caused a significant increase in EPA and DPA but not DHA in red blood cells, serum and liver suggesting elongation and desaturation of the supplemented EPA. When the dietary supplement contained algal oil of DHA alone liver content of DPA and DHA significantly increased suggesting retro conversion of a portion of the DHA to DPA. This was not observed in the serum or when DHA alone was provided by the ethyl ester. There was generally an increase in the unsaturation index in the serum and liver compared to control and/or untreated diabetic rats when diabetic rats were treated with menhaden oil, krill oil or algal oils of EPA, DHA, or EPA + DHA. The unsaturation index was not statistically changed when EPA, DHA or EPA + DHA were provided as an ethyl ester. To compensate for the increase in omega-3 PUFAs in serum and liver following treatment of diabetic rats with dietary sources of omega-3 PUFAs there was a general decrease in levels of linolenic and arachidonic acids, both omega-6 PUFAs. This resulted in a lower omega-6/omega-3 PUFA ratio. Lowering this ratio is associated with a lower risk of metabolic syndrome and prevention/treatment for non-alcoholic fatty liver disease [25,26]. These data demonstrate that dietary supplementation with a natural source of omega-3 PUFAs vs. a manufactured source provide similar changes in the fatty acid composition of red blood cells, serum and tissue. This also appears to be independent on whether the omega-3 PUFAs supplement is triglyceride based as in fish oil or phospholipid based as in krill oil [27,28].

There has been a long history of studies proclaiming fish oil derived omega-3 PUFAs as a nutritional benefit for a multitude of diseases [29]. However, this statement is not without controversy [29]. Currently, it is generally accepted that omega-3 PUFAs are beneficial in the treatment of cardiovascular disease even though there have been many clinical trials and meta-analyses that have reported otherwise [30,31,32]. One consistent problem with these past studies has been the lack of determining the omega-3 index or levels of omega-3 PUFAs in circulation pre- and post-treatment. Many of these studies may have used dosage of omega-3 PUFAs that was too low to achieve a “healthy” range. In the REDUCE-IT study it was reported that treatment with EPA alone in the form of an ethyl ester, icosapent ethyl at 4 g/day, was highly effective as a treatment of heart disease and is considered a promising therapy for atherosclerotic cardiovascular disease [33,34,35]. 

We had previously reported that menhaden oil was a beneficial treatment of obese or type 1 or type 2 diabetic rat models of neural and vascular impairment [7,8,36]. In this study we found that the neural endpoints impacted by type 2 diabetes as well as vascular dysfunction of epineurial arterioles to acetylcholine and CGRP were also significantly improved when the diabetic rats were treated, using an early or late intervention protocol, with krill oil or EPA + DHA derived from algae or ethyl esters. When diabetic rats were treated with algal or ethyl ester of EPA or DHA alone the effect was reduced and treatment with DHA alone was observed to be more beneficial than EPA alone. Our study demonstrated that fish oil is not the only source of omega-3 PUFAs that can improve DPN and unlike cardiovascular disease the combination of EPA + DHA was more efficacious than EPA alone. However, in the REDUCE-IT study the effect of EPA + DHA or DHA alone as an ethyl ester was not studied.

Lewis et al. reported that omega-3 PUFAs supplementation using seal oil was associated with an increase in corneal nerve fiber length in subjects with type 1 diabetes [37]. Studies from this laboratory have also reported that low plasma levels of DHA were associated with diabetic distal symmetric polyneuropathy and higher baseline omega-3 PUFAs were associated with greater nerve regeneration [38]. Previously we had reported that vascular dysfunction of epineurial precedes the development of deficits in nerve conduction velocity [39]. These arterioles are resistance size vessels, and it has been reported that use of fish oil supplementation and higher plasma levels of omega-3 PUFA, especially DHA, was associated with lower risks of macro- and micro-vascular complications in individuals with type 2 diabetes [40]. Omega-3 PUFAs supplements have also acquired interest as a potential treatment for peripheral neuropathies related to chemotherapy and radiotherapy [41,42].

Enrichment with krill oil has been shown to be therapeutic on obesity-induced metabolic syndrome in mice and cardiovascular risk factors in patients with type 2 diabetes [43,44,45]. However, no pre-clinical or clinical studies are available on its potential benefits for peripheral neuropathy. Likewise, little information is available of the potential benefits of omega-3 PUFAs derived from algae or pharmaceutical-derived ethyl esters on peripheral neuropathy. Algae are the primary source of omega-3 PUFAs in the food chain for marine fish and mammals and different species exist that produce primarily EPA or DHA or the combination of EPA + DHA [46,47]. Algal DHA-rich oil has been shown to improve some markers of cardiometabolic risk and decreased plasma levels of triglycerides [48]. Interestingly, we observed a similar effect on serum triglyceride levels in this study.

An unresolved question is whether EPA or DHA alone are beneficial omega-3 fatty acid supplements or does the combination of EPA + DHA provide the best outcome. Our results would suggest that for the best results on DPN the combination of EPA + DHA is required. In the REDUCE-IT study icosapent ethyl was found to reduce the risk of ischemic events [33]. However, these studies failed to examine the effect of the ethyl ester of DHA. Van et al. demonstrated that esterification of DHA enhanced its transport to the brain [47]. This study did not examine whether esterification of EPA was beneficial to transport. A study by Joardar and Chakraborty, concluded that EPA and DHA have differential effects on membrane dynamics [49]. Additional studies will be needed to address this complicated issue including the individual role of omega-3 PUFA metabolites [50,51,52,53]. Little is known about the most efficacious dose of omega-3 fatty acids to take as a supplement to maintain a healthy omega-3 index and how other drugs such as non-steroid anti-inflammatory agents act in combination with fish oil supplements to promote formation of resolvins and neuroprotection. This study is in progress NCT05169060.

A limitation of these study results is that the algal oil EPA was from a different supplier and was not as refined as the oils from DSM Nutritional Products or krill oil. Nonetheless, when incorporated into the diet the fatty acid composition of serum, red blood cell membranes, and liver were compatible to the other omega-3 fatty acids sources and results on DPN, and diabetes-induced vascular dysfunction were similar between EPA alone derived from algal oil or as an ethyl ester.

## 5. Conclusions

These studies demonstrated that omega-3 PUFAs from sources other than fish oil are effective in slowing the progression and reversing DPN. Sources containing the combination of EPA and DHA were more effective than either EPA or DHA alone. 

## Figures and Tables

**Figure 1 biomedicines-13-01607-f001:**
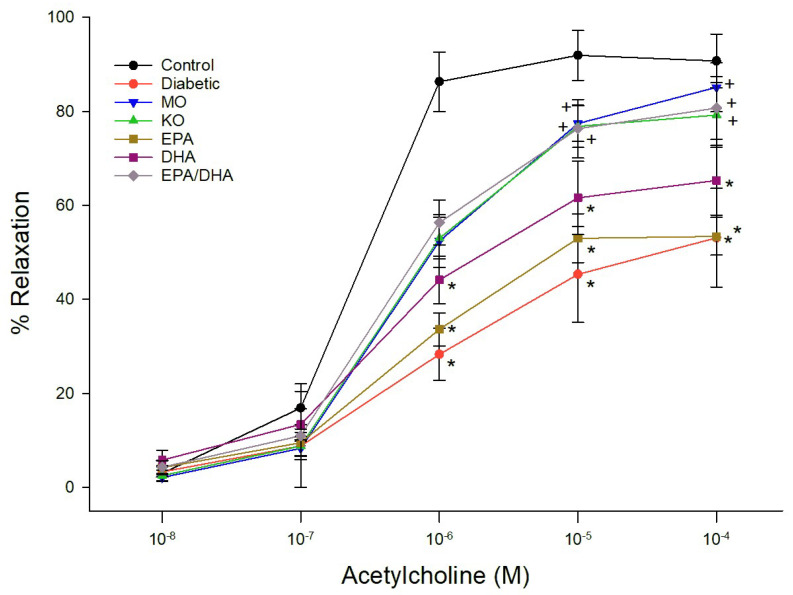
Effect of dietary intervention with menhaden oil, krill oil, and oils from algae that produce EPA, DHA or the combination of EPA + DHA on vascular relaxation by acetylcholine in epineurial arterioles of the sciatic nerve in diabetic Sprague-Dawley rats. Pressurized arterioles (40 mm Hg and ranging from 60–100 µm luminal diameters) were constricted with phenylephrine (30–50%) and incremental doses of acetylcholine were added to the bathing solution while recording steady state vessel diameter. The number of rats in each group ranged from 8–9. Data are presented as the mean of % relaxation ± S.E.M. * *p* < 0.05 compared to control rats. ^+^
*p* < 0.05 compared to diabetic rats.

**Figure 2 biomedicines-13-01607-f002:**
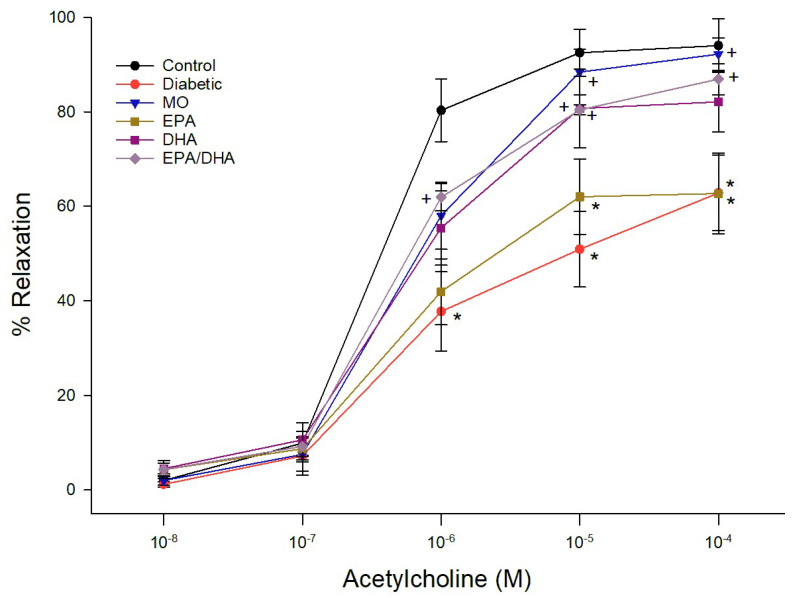
Effect of dietary intervention with menhaden oil, and ethyl esters of EPA, DHA or the combination of EPA + DHA on vascular relaxation by acetylcholine in epineurial arterioles of the sciatic nerve in diabetic Sprague-Dawley rats. Pressurized arterioles (40 mm Hg and ranging from 60–100 µm luminal diameters) were constricted with phenylephrine (30–50%) and incremental doses of acetylcholine were added to the bathing solution while recording steady state vessel diameter. The number of rats in each group ranged from 8–9. Data are presented as the mean of % relaxation ± S.E.M. * *p* < 0.05 compared to control rats. ^+^
*p* < 0.05 compared to diabetic rats.

**Figure 3 biomedicines-13-01607-f003:**
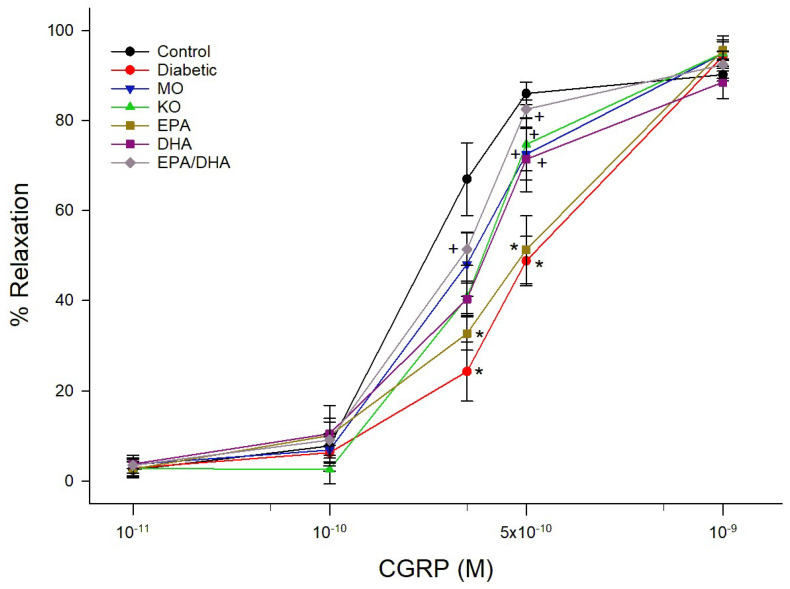
Effect of dietary intervention with menhaden oil, krill oil, and oils from algae that produce EPA, DHA or the combination of EPA + DHA on vascular relaxation by CGRP in epineurial arterioles of the sciatic nerve in diabetic Sprague-Dawley rats. Pressurized arterioles (40 mm Hg and ranging from 60–100 µm luminal diameters) were constricted with phenylephrine (30–50%) and incremental doses of CGRP were added to the bathing solution while recording steady state vessel diameter. The number of rats in each group ranged from 8–9. Data are presented as the mean of % relaxation ± S.E.M. * *p* < 0.05 compared to control rats. ^+^
*p* < 0.05 compared to diabetic rats.

**Figure 4 biomedicines-13-01607-f004:**
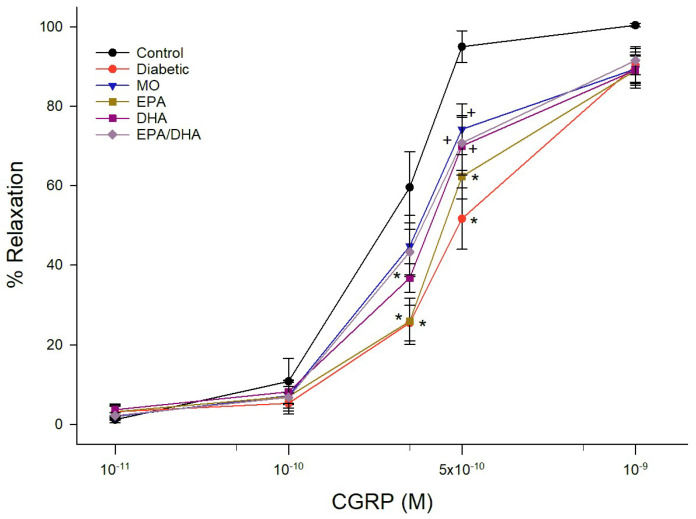
Effect of dietary intervention with menhaden oil, and ethyl esters of EPA, DHA or the combination of EPA + DHA on vascular relaxation by CGRP in epineurial arterioles of the sciatic nerve in diabetic Sprague-Dawley rats. Pressurized arterioles (40 mm Hg and ranging from 60–100 µm luminal diameters) were constricted with phenylephrine (30–50%) and incremental doses of CGRP were added to the bathing solution while recording steady state vessel diameter. The number of rats in each group ranged from 8–9. Data are presented as the mean of % relaxation ± S.E.M. * *p* < 0.05 compared to control rats. ^+^
*p* < 0.05 compared to diabetic rats.

**Table 1 biomedicines-13-01607-t001:** Fatty Acid % Composition of Red Blood Cell Membranes and Summation of EPA, DPA and DHA after 12 Weeks of Treatment.

Diet	16:0	18:0	18:1	18:2	20:4	20:5	22:5	22:6	∑ EPA, DPA, DHA
Control	28.3 ± 2.7	16.2 ± 1.1	8.6 ± 0.6	7.8 ± 0.9	20.2 ± 1.4	<1	1.4 ± 0.3	1.2 ± 0.2	2.7 ± 0.1
Diabetic	25.6 ± 1.9	18.4 ± 1.5	9.6 ± 0.7	9.4 ± 0.9	20.9 ± 1.7	<1	1.3 ± 0.1	1.7 ± 0.3	3.4 ± 0.2
Diabetic + Menhaden oil	23.2 ± 1.2	17.5 ± 0.8	9.1 ± 0.5	8.7 ± 0.6	14.2 ± 0.9	4.8 ± 0.3	3.9 ± 0.1	4.0 ± 0.1	12.7 ± 0.6 ^a,b^
Diabetic + Krill oil	29.9 ± 1.7	14.8 ± 1.0	9.2 ± 0.6	8.9 ± 0.6	10.0 ± 0.8	6.8 ± 0.4	3.4 ± 0.2	4.1 ± 0.2	14.3 ± 0.6 ^a,b^
Diabetic + Algal oil EPA	22.9 ± 1.4	14.8 ± 1.4	6.2 ± 0.3	8.8 ± 0.6	15.0 ± 1.0	4.0 ± 0.2	4.8 ± 0.7	<1	9.4 ± 0.5 ^a,b^
Diabetic + Algal oil DHA	25.2 ± 1.7	12.9 ± 0.8	6.1 ± 0.3	7.9 ± 0.6	16.5 ± 1.3	<1	1.0 ± 0.2	7.1 ± 0.3	8.9 ± 0.2 ^a,b^
Diabetic + Algal oil EPA/DHA	26.1 ± 1.1	16.6 ± 1.1	8.3 ± 0.4	7.8 ± 0.4	12.9 ± 0.9	5.0 ± 0.3	2.0 ± 0.3	6.0 ± 0.3	13.0 ± 0.4 ^a,b^
Diabetic + Ethyl Ester EPA	23.9 ± 0.6	15.8 ± 0.7	8.2 ± 0.5	10.1 ± 0.9	17.2 ± 1.0	4.6 ± 0.2	4.0 ± 0.7	<1	8.8 ± 0.2 ^a,b^
Diabetic + Ethyl Ester DHA	27.5 ± 1.6	15.9 ± 1.0	7.7 ± 0.4	10.2 ± 0.7	14.0 ± 1.0	<1	2.4 ± 0.1	5.6 ± 0.2	8.8 ± 0.3 ^a,b^
Diabetic + Ethyl Ester EPA/DHA	23.7 ± 1.8	16.1 ± 1.3	8.8 ± 0.7	9.0 ± 0.4	14.6 ± 1.0	2.6 ± 0.2	6.8 ± 0.2	4.8 ± 0.4	14.2 ± 0.6 ^a,b^

Data are presented as the mean of three determinations ± S.E.M. * Omega-3 index equals summation of eicosapentaenoic acid (EPA, 20:5), docosapentaenoic acid (DPA, 22:5) and docosahexaenoic acid (DHA, 22:6). ^a^
*p* < 0.05 vs. control rats; ^b^
*p* < 0.05 vs. diabetic rats.

**Table 2 biomedicines-13-01607-t002:** Effect of Early Intervention of Omega-3 Polyunsaturated Fatty Acid derived from Menhaden oil, Krill oil, or Algal oils enriched in EPA, DHA or EPA + DHA on Peripheral Neuropathy Related Endpoints in Type 2 Diabetic Sprague Dawley Rats.

Conditions	MNCV (m/s)	SNCV (m/s)	Heat Sensitivity (s)	IENF (Profiles/mm)	Cornea Nerve Fiber Length (mm/mm^2^)	Cornea Sensitivity (cm)	Cornea Sensitivity (AUC)
Control	53.1 ± 1.5	39.2 ± 0.7	11.6 ± 0.7	24.1 ± 0.9	7.5 ± 0.4	5.94 ± 0.04	27 ± 9
Diabetic	40.2 ± 1.2 ^a^	32.5 ± 0.9 ^a^	20.6 ± 1.7 ^a^	14.0 ± 0.6 ^a^	3.2 ± 0.2 ^a^	4.66 ± 0.21 ^a^	118 ± 8 ^a^
Diabetic + MO	52.2 ± 1.4 ^b^	37.2 ± 0.6 ^b^	12.7 ± 0.8 ^b^	17.2 ± 0.6 ^a,b^	6.9 ± 0.5 ^b^	5.84 ± 0.08 ^b^	34 ± 9 ^b^
Diabetic + KO	49.6 ± 1.6 ^b^	37.2 ± 1.1 ^b^	13.1 ± 0.9 ^b^	18.4 ± 0.3 ^a,b^	6.9 ± 0.5 ^b^	5.72 ± 0.07 ^b^	47 ± 13 ^b^
Diabetic + EPA	44.6 ± 1.1	36.1 ± 0.9	11.1 ± 1.2 ^b^	19.5 ± 1.0 ^a,b^	4.4 ± 0.2 ^a^	5.72 ± 0.10 ^b^	70 ± 15
Diabetic + DHA	49.7 ± 1.9 ^b^	37.6 ± 0.5 ^b^	12.2 ± 0.8 ^b^	19.1 ± 0.8 ^a,b^	6.4 ± 0.3 ^b^	5.88 ± 0.07 ^b^	64 ± 7 ^b^
Diabetic + EPA + DHA	53.2 ± 1.8 ^b^	38.1 ± 0.7 ^b^	12.1 ± 0.9 ^b^	21.8 ± 0.6 ^b^	6.3 ± 0.5 ^b^	5.94 ± 0.04 ^b^	65 ± 4 ^b^

Data are presented as the mean ± S.E.M. *n* = 8–9. MNCV: motor nerve conduction velocity, SNCV: sensory nerve conduction velocity, IENF: intraepidermal nerve fiber density, MO: menhaden oil, KO: krill oil, AO: algal oil, EPA: eicosapentaenoic acid, DHA: docosahexaenoic acid. The number of animals for each experimental group was 8–9. ^a^: *p* < 0.05 vs. control rats; ^b^: *p* < 0.05 vs. diabetic rats.

**Table 3 biomedicines-13-01607-t003:** Effect of Late Intervention of Omega-3 Polyunsaturated Fatty Acid derived from Menhaden oil, Krill oil, or Algal oils enriched in EPA, DHA or EPA + DHA on Peripheral Neuropathy Related Endpoints in Type 2 Diabetic Sprague Dawley Rats.

Conditions	MNCV (m/s)	SNCV (m/s)	Heat Sensitivity (s)	IENF (Profiles/mm)	Corneal Nerve Fiber Length (mm/mm^2^)	Cornea Sensitivity (cm)	Cornea Sensitivity (AUC)
Control	56.9 ± 1.8	39.0 ± 0.4	12.2 ± 0.7	22.5 ± 0.9	7.5 ± 0.3	5.88 ± 0.06	54 ± 4
Diabetic	40.0 ± 1.0 ^a^	32.8 ± 0.8 ^a^	20.8 ± 1.2 ^a^	15.0 ± 0.2 ^a^	3.2 ± 0.3 ^a^	4.98 ± 0.15 ^a^	155 ± 15 ^a^
Diabetic + MO	50.5 ± 1.0 ^b^	40.1 ± 0.7 ^b^	14.4 ± 0.8 ^b^	19.0 ± 0.4 ^a,b^	6.8 ± 0.3 ^b^	5.80 ± 0.07 ^b^	58 ± 10
Diabetic + KO	52.3 ± 1.9 ^b^	38.0 ± 0.9 ^b^	12.3 ± 0.8 ^b^	17.8 ± 0.5 ^a,b^	5.7 ± 0.6 ^b^	5.80 ± 0.07 ^b^	63 ± 7
Diabetic + EPA	45.8 ± 2.1 ^a^	36.2 ± 0.6 ^a^	12.1 ± 0.6 ^b^	21.0 ± 0.6 ^b^	4.6 ± 0.4 ^a^	5.57 ± 0.08 ^b^	85 ± 12
Diabetic + DHA	50.9 ± 1.8 ^b^	38.7 ± 0.6 ^b^	12.7 ± 0.5 ^b^	20.9 ± 0.9 ^b^	6.4 ± 0.6 ^b^	5.81 ± 0.06 ^b^	61 ± 8
Diabetic + EPA + DHA	49.4 ± 1.1 ^b^	39.5 ± 0.8 ^b^	12.3 ± 0.5 ^b^	21.6 ± 0.7 ^b^	6.5 ± 0.4 ^b^	5.82 ± 0.06 ^b^	58 ± 12

Data are presented as the mean ± S.E.M. *n* = 8–9. MNCV: motor nerve conduction velocity, SNCV: sensory nerve conduction velocity, IENF: intraepidermal nerve fiber density, MO: menhaden oil, KO: krill oil, AO: algal oil, EPA: eicosapentaenoic acid, DHA: docosahexaenoic acid. The number of animals for each experimental group was 8–9. ^a^: *p* < 0.05 vs. control rats; ^b^: *p* < 0.05 vs. diabetic rats.

**Table 4 biomedicines-13-01607-t004:** Effect of Early Intervention of Omega-3 Polyunsaturated Fatty Acid derived from Menhaden oil, or Ethyl Esters of EPA, DHA or EPA + DHA on Peripheral Neuropathy Related Endpoints in Type 2 Diabetic Sprague Dawley Rats.

Conditions	MNCV (m/s)	SNCV (m/s)	Heat Sensitivity (s)	IENF (Profiles/mm)	Corneal Nerve Fiber Length (mm/mm^2^)	Cornea Sensitivity (cm)	Cornea Sensitivity (AUC)
Control	56.1 ± 2.4	38.8 ± 0.6	12.3 ± 0.3	24.1 ± 0.9	7.5 ± 0.3	5.86 ± 0.05	35 ± 5
Diabetic	42.6 ± 1.2 ^a^	32.9 ± 1.0 ^a^	22.6 ± 0.8 ^a^	14.0 ± 0.6 ^a^	3.2 ± 0.2 ^a^	4.53 ± 0.15 ^a^	150 ± 4 ^a^
Diabetic + MO	52.6 ± 1.4	39.1 ± 0.6 ^b^	12.8 ± 0.8 ^b^	17.2 ± 0.6 ^a,b^	6.4 ± 0.3 ^b^	5.84 ± 0.06 ^b^	33 ± 9 ^b^
Diabetic + EPA	43.7 ± 2.2 ^a^	36.4 ± 1.2	11.9 ± 0.7 ^b^	18.6 ± 0.8 ^a,b^	4.2 ± 0.7 ^a^	5.53 ± 0.12 ^b^	52 ± 14 ^b^
Diabetic + DHA	53.4 ± 1.7	39.0 ± 0.7 ^b^	13.5 ± 0.7 ^b^	18.9 ± 0.5 ^a,b^	5.5 ± 0.6 ^a,b^	5.83 ± 0.08 ^b^	45 ± 9 ^b^
Diabetic + EPA + DHA	51.4 ± 1.5	38.7 ± 0.7 ^b^	12.9 ± 0.7 ^b^	20.9 ± 1.0 ^b^	6.2 ± 0.2 ^b^	5.84 ± 0.08 ^b^	40 ± 11 ^b^

Data are presented as the mean ± S.E.M. *n* = 8–9. MNCV: motor nerve conduction velocity, SNCV: sensory nerve conduction velocity, IENF: intraepidermal nerve fiber density, MO: menhaden oil, or Ethyl Esters, EPA: eicosapentaenoic acid, DHA: docosahexaenoic acid. The number of animals for each experimental group was 8–9. ^a^: *p* < 0.05 vs. control rats; ^b^: *p* < 0.05 vs. diabetic rats.

**Table 5 biomedicines-13-01607-t005:** Effect of Late Intervention of Omega-3 Polyunsaturated Fatty Acid derived from Menhaden oil, or Ethyl Esters of EPA, DHA or EPA + DHA on Peripheral Neuropathy Related Endpoints in Type 2 Diabetic Sprague Dawley Rats.

Conditions	MNCV (m/s)	SNCV (m/s)	Heat Sensitivity (s)	IENF (Profiles/mm)	Corneal Nerve Fiber Length (mm/mm^2^)	Cornea Sensitivity (cm)	Cornea Sensitivity (AUC)
Control	55.2 ± 1.8	40.0 ± 1.0	12.2 ± 0.4	22.5 ± 0.9	8.5 ± 0.4	5.88 ± 0.06	47 ± 9
Diabetic	39.7 ± 1.7 ^a^	32.3 ± 1.4 ^a^	21.4 ± 1.2 ^a^	15.0 ± 0.2 ^a^	3.4 ± 0.3 ^a^	4.92 ± 0.14 ^a^	116 ± 8 ^a^
Diabetic + MO	52.8 ± 1.7 ^b^	37.5 ± 0.3 ^b^	12.0 ± 0.5 ^b^	19.0 ± 0.3 ^a,b^	6.3 ± 0.4 ^a,b^	5.83 ± 0.07 ^b^	28 ± 6 ^b^
Diabetic + EPA	46.8 ± 1.7 ^a,b^	36.0 ± 1.1	17.8 ± 1.5 ^a^	21.2 ± 0.2 ^b^	4.4 ± 0.4 ^a^	5.60 ± 0.10 ^b^	77 ± 18
Diabetic + DHA	50.8 ± 0.9 ^b^	37.5 ± 0.3 ^b^	14.8 ± 0.7 ^b^	19.1 ± 0.4 ^b^	7.6 ± 0.5 ^b^	5.78 ± 0.07 ^b^	45 ± 11 ^b^
Diabetic + EPA+ DHA	50.5 ± 1.3 ^b^	39.9 ± 1.5 ^b^	12.5 ± 0.8 ^b^	22.0 ± 0.7 ^b^	7.2 ± 0.4 ^b^	5.79 ± 0.09 ^b^	57 ± 14 ^b^

Data are presented as the mean ± S.E.M. *n* = 8–9. MNCV: motor nerve conduction velocity, SNCV: sensory nerve conduction velocity, IENF: intraepidermal nerve fiber density, MO: menhaden oil, or Ethyl Esters, EPA: eicosapentaenoic acid, DHA: docosahexaenoic acid. The number of animals for each experimental group was 8–9. ^a^: *p* < 0.05 vs. control rats; ^b^: *p* < 0.05 vs. diabetic rats.

## Data Availability

The datasets generated for this study will be made available on request to the corresponding author and authorization by the office of the OIG since this study was supported by Veterans Affairs.

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
