# Peer review of "Omega-3 Polyunsaturated Fatty Acids (PUFAs) and Diabetic Peripheral Neuropathy: A Pre-Clinical Study Examining the Effect of Omega-3 PUFAs from Fish Oil, Krill Oil, Algae or Pharmaceutical-Derived Ethyl Esters Using Type 2 Diabetic Rats"

_biomedicines, 2025, doi:10.3390/biomedicines13071607_

Round 1
Reviewer 1 Report
Comments and Suggestions for Authors
Dear Erudite Editor and Esteemed Authors of the manuscript biomedicines-3670238, thank you for inviting me to review it. The present work is intriguing; however, I would like to address a few concerns with the Esteemed Authors before the Erudite Editor decides.
MAJOR POINTS
- The abstract lacks references to the results discussed in your manuscript. You might decrease the lengthy introduction and methods sections and focus on the results in the proper section, including significance levels and the most prominent results. You may also modify the conclusions section to delve into the potential clinical implications of your findings and future research endeavors.
- The introduction section lacks sufficient background on diabetic neuropathy. Try to include text about its pathogenesis and the rationale behind using omega-3 to counteract this complication. Include a figure delving into the justification for the conduct of this study. Compare your results to those of other published studies and justify the prominent importance of your findings.
- Please use numbered subsections in your methodology section. Additionally, your methodology lacks essential information, including animal security data and sufficient background information regarding the study protocol, including a more in-depth exploration of the doses used for omega-3 acids. Also, include a figure about your study design in your methodology section.
- Your discussion section doesn't seem to have enough information about the future research endeavors and possible clinical implications of your findings. Please include dedicated paragraphs delving into these aspects of medical research.
- In your conclusion, please include an illustration representing your main findings, the clinical implications of your study, and the future research endeavors based on your results.
- The anti-inflammatory and antioxidant properties of Icosapent, a critical omega-3, must be discussed briefly in your manuscript. See the recently published manuscript in this regard, like https://doi.org/10.3390/ph18040601 and others.
- Your similarity report is counting up to 41%. This is unacceptable. To ensure a swift publication, you must lower this percentage to less than 15%.
- Your reference style is completely wrong based on MDPI’s standards. Please review MDPI’s policies for in-text citations and references list to provide aligned references.
Author Response
Response to Reviewer 1:
- The abstract lacks references to the results discussed in your manuscript. You might decrease the lengthy introduction and methods sections and focus on the results in the proper section, including significance levels and the most prominent results. You may also modify the conclusions section to delve into the potential clinical implications of your findings and future research endeavors.
Response: I appreciate the reviewer’s viewpoint. This was a complicated study using different sources of reagents to obtain the omega-3 fatty acids in their different forms. After much consideration I do not think that that the introduction and results sections of the abstract were lengthy because it was needed to define the study and without it the reader would not know what the multiple results referred to. Therefore, I respectively would like to leave the abstract in its present form.
- The introduction section lacks sufficient background on diabetic neuropathy. Try to include text about its pathogenesis and the rationale behind using omega-3 to counteract this complication. Include a figure delving into the justification for the conduct of this study. Compare your results to those of other published studies and justify the prominent importance of your findings.
Response: An additional paragraph has been included to address this concern.
- Please use numbered subsections in your methodology section. Additionally, your methodology lacks essential information, including animal security data and sufficient background information regarding the study protocol, including a more in-depth exploration of the doses used for omega-3 acids. Also, include a figure about your study design in your methodology section.
Response: Providing the methods in significant detail would be an easy cut and paste process but would add 2-3 pages to an already lengthy manuscript. Each of these methods have been used extensively by my laboratory for many years and in the case of cornea nerve imaging in rats and the vascular studies were developed by my laboratory. The references provided give directions for each of the endpoint studies in detail.
- Your discussion section doesn't seem to have enough information about the future research endeavors and possible clinical implications of your findings. Please include dedicated paragraphs delving into these aspects of medical research.
Response: This important issue has been expanded in the end of the discussion section as well as the conclusion section.
- In your conclusion, please include an illustration representing your main findings, the clinical implications of your study, and the future research endeavors based on your results.
Response: I would prefer to leave such graphics to review articles. There are many endpoints that were examined and are not easy to represent in a graphic.
- The anti-inflammatory and antioxidant properties of Icosapent, a critical omega-3, must be discussed briefly in your manuscript. See the recently published manuscript in this regard, like https://doi.org/10.3390/ph18040601 and others.
Response: Thank you for this reference and we have included it in the discussion on this subject. Meta-analyses of past studies relating to the effect of omega-3 fatty acids on cardiovascular disease can be misleading. There have been many problems regarding past studies of omega-3 fatty acids in humans but the most significant are that almost all of the studies performed failed to examine omega-3 fatty acid levels at baseline and during treatment. Therefore it is not possible to determine if the dosing was sufficient. The REDUCE-IT study used 4 grams of Icosapent and observed beneficial effects, but that study did not measure the levels of EPA, DPA, or DHA in their patients. Studies have not examined the potential benefit of other ethyl ester omega-3 fatty acids such as DHA or the combination of EPA and DHA. Most studies in the literature used 1 gram of omega-3 fatty acid supplement and likely never achieved a pharmaceutical range.
- Your similarity report is counting up to 41%. This is unacceptable. To ensure a swift publication, you must lower this percentage to less than 15%.
Response: This was surprising, and we have attempted to address it but most of the redundancy in terms accumulate from similarity values of 1 or 2% adding up to 41%. There was no redundant use of terms from any 1 paper.
- Your reference style is completely wrong based on MDPI’s standards. Please review MDPI’s policies for in-text citations and references list to provide aligned references.
Response: This has been corrected.
Reviewer 2 Report
Comments and Suggestions for Authors
Review of “Is Fish Oil the Only Efficacious Source of Omega-3 Polyunsaturated Fatty Acids for the Treatment of Diabetic Peripheral Neuropathy? A Pre-Clinical Study Using Type 2 Diabetic Rats”
I found this study is well designed and made, although methodic section is very rudimental and need to be improved before accepted. There are some questions about comparison with passive control group in contrast to diabetic one, which must be answered or changed. Overall, there is a good scientific problem chosen, resolved with appropriate methods, but some minor labor in presentation of results still needed. I recommend accepting after minor revision.
- Didn’t found in Methods section which method(s) used to measure concentration of fatty acids in the red blood cells, serum and liver. Need to be explained in the methods. Also lack the information on used endpoints measures – need to be explained in details. All studies are more reproducible when methods section goes far away then one paragraph…
- Authors choice of information mode was tables, but now it is more visual mode demand – can you please provide plots showing you results, it is much more easily understood.
- Supplemental Table 2 provide information on effects of oil intervention on diabetic rats, but most of comparison made with control rats (marked as a), but must be made with diabetic rats without intervention (marked b). Please explain why.
- Same for Supplemental Table 3. What about multiple comparison correction, like Bonferroni correction – was it made or not? Why some of data was compared with native controls, and part was compared with diabetic rats, is it not p-hacking?
- Same for upcoming Supplemental Tables.
- The same question for Fig 1 - why the comparison made with control rats, not diabetic one?
- First paragraphs of discussion can be moved to the Intro section easily and with benefit to readers.
- From abstract: “Primary endpoints were motor and sensory nerve conduction velocity, intraepidermal and cornea nerve fiber density, thermal nociception, cornea sensitivity, and vascular reactivity of epineurial arterioles of the sciatic nerve.” – I didn’t find some of mentioned endpoints/measures in provided figures – can you please revise it to make actual? Where is data on density and nociception, for example? Figures is better than tables, again, for comprehension – tables can go to supplementary for pedants.
Author Response
Response to Reviewer 2:
Didn’t found in Methods section which method(s) used to measure concentration of fatty acids in the red blood cells, serum and liver. Need to be explained in the methods. Also lack the information on used endpoints measures – need to be explained in details. All studies are more reproducible when methods section goes far away then one paragraph…
Response: These methods have been provided in more detail.
Authors choice of information mode was tables, but now it is more visual mode demand – can you please provide plots showing you results, it is much more easily understood.
Response: I agree with the reviewer and would like to comply with the request but this would likely require 12-16 figures, which is likely beyond standard for the journal.
Supplemental Table 2 provide information on effects of oil intervention on diabetic rats, but most of comparison made with control rats (marked as a), but must be made with diabetic rats without intervention (marked b). Please explain why.
Response: The data is compared to untreated diabetic rats (superscript b). There was some effect of the different treatments on serum free fatty acid, triglyceride or cholesterol levels.
Same for Supplemental Table 3. What about multiple comparison correction, like Bonferroni correction – was it made or not? Why some of data was compared with native controls, and part was compared with diabetic rats, is it not p-hacking?
Response: We performed multiple comparison correction but did not show the results (there were only a few) since we thought it would distract from the main points.
Same for upcoming Supplemental Tables.
The same question for Fig 1 - why the comparison made with control rats, not diabetic one?
Response: This has been corrected
First paragraphs of discussion can be moved to the Intro section easily and with benefit to readers.
Response: I respectively disagree. The introduction focuses primarily on DPN and lack of success in treatment. The first paragraph in the discussion I think sets up this section for focus on omega-3 fatty acids.
From abstract: “Primary endpoints were motor and sensory nerve conduction velocity, intraepidermal and cornea nerve fiber density, thermal nociception, cornea sensitivity, and vascular reactivity of epineurial arterioles of the sciatic nerve.” – I didn’t find some of mentioned endpoints/measures in provided figures – can you please revise it to make actual? Where is data on density and nociception, for example? Figures is better than tables, again, for comprehension – tables can go to supplementary for pedants.
Response: This has been addressed except for replacing tables with figures, which would result in 12-16 additional figures.
Reviewer 3 Report
Comments and Suggestions for Authors
In this research article entitled “Is Fish Oil the Only Efficacious Source of Omega-3 Polyunsaturated Fatty Acids for the Treatment of Diabetic Peripheral Neuropathy? A Pre-Clinical Study Using Type 2 Diabetic Rats.” authored by Davidson et al., authors tried to answer the question that if Fish Oil is the Only Efficacious Source of Omega-3 Polyunsaturated Fatty Acids for the Treatment of Diabetic Peripheral Neuropathy. Although it appears to be an interesting work however, it requires extensive revision:
- The title of this manuscript is confusing. It should be revised with a straight forward message.
- Abstract in the current form is not acceptable and it should be revised to make it clearer and easier understandable.
- It has been well drafted, but language quality is not satisfactory. Flow information is not observed.
- The introduction portion must be revised.
- Figures and graphs are difficult to understand. Colourful graphs are suggested to be added
- It is suggested to rephrase the long and difficult sentences into smaller and understandable ones.
- Referenced style should be corrected.
- Materials and methods section required thorough revision.
- How DPN was induced? How was it confirmed? No info is provided
- What is Diabetes peripheral neuropathy?
Author Response
Response to Reviewer 3:
- The title of this manuscript is confusing. It should be revised with a straight forward message.
Response: The title has been changed.
- Abstract in the current form is not acceptable and it should be revised to make it clearer and easier understandable.
Response: Given the word limitations it is not possible to add a large amount of result information of the abstract. The other reviewers did not object of the abstract in its present form.
- It has been well drafted, but language quality is not satisfactory. Flow information is not observed.
Response: This has been addressed.
- The introduction portion must be revised.
Response: This has been addressed.
- Figures and graphs are difficult to understand. Colourful graphs are suggested to be added
Response: The different ticks for the different conditions have been enlarged so that the data is easier to follow.
- It is suggested to rephrase the long and difficult sentences into smaller and understandable ones.
Response: This has been addressed.
- Referenced style should be corrected.
Response: This has been corrected.
- Materials and methods section required thorough revision.
Response: This has been revised.
- How DPN was induced? How was it confirmed? No info is provided
Response: This has been clarified in the Methods section.
- What is Diabetes peripheral neuropathy?
Response: This has been addressed in the Introduction section.
Reviewer 4 Report
Comments and Suggestions for Authors
The article is well-organized and methodologically rigorous preclinical study comparing the efficacy of various dietary sources of omega-3 polyunsaturated fatty acids (PUFAs) as therapeutic agents for diabetic peripheral neuropathy (DPN) in a type 2 diabetic rat model. The researchers show that along with fish oil, krill oil, algal oils, and ethyl esters of EPA and DHA are also effective in alleviating neural and vascular complications of diabetes. For me the notable point is that the EPA + DHA combination, regardless of source, invariably had better therapeutic effects than monocomponent treatments. Utilizing multiple time points of intervention ( like early vs. late) maximizes the translational value of the overall study.
The manuscript is, however, seriously impaired by 41% plagiarism, which brings into question ethical integrity in terms of originality. So, please take care of this point.
Although experimental design and findings are intriguing; proper citation and rewriting of copied passages are required by the authors to uphold scientific integrity.
Generally, the research provides worthwhile comparative observations on alternative sources of omega-3, which is opportune against sustainability and environmental concerns. Assuming the plagiarism problem is fixed, this work has potential for publication in a high-impact biomedical journal.
Please improv the manuscript before publication by:
Minimizing plagiarism
Explaining variability in oil refining and its potential influence on outcomes.
Elaborate on translational significance of findings for informing human clinical trials.
Incorporate power analysis to support sample size. Presently, it is missing in the study.
Don't forget to provide raw full datasets in supplementary documents.
Author Response
Response to Reviewer 4:
The manuscript is, however, seriously impaired by 41% plagiarism, which brings into question ethical integrity in terms of originality. So, please take care of this point. Response: I assure the reviewer that everything presented and discussed was original work and properly cited when pertaining to other studies. There was no plagiarism. We have edited the contents to attempt to reduce the “word similarity” that in itself is it plagiarism. In the case of this paper there are multiple claims of word similarity to other papers, primarily our own, of 1-2% and over a couple dozen papers a match of a few words not complete statements that were not cited is not plagiarism. We have tried to reduce this.
Although experimental design and findings are intriguing; proper citation and rewriting of copied passages are required by the authors to uphold scientific integrity. Response: This has been done.
Generally, the research provides worthwhile comparative observations on alternative sources of omega-3, which is opportune against sustainability and environmental concerns. Assuming the plagiarism problem is fixed, this work has potential for publication in a high-impact biomedical journal. Response: Thank you for the comment.
Please improv the manuscript before publication by:
Minimizing plagiarism Response: This has been addressed.
Explaining variability in oil refining and its potential influence on outcomes. Response: This has been commented.
Elaborate on translational significance of findings for informing human clinical trials. Response: This has been addressed.
Incorporate power analysis to support sample size. Presently, it is missing in the study. Response: This has been addressed.
Round 2
Reviewer 1 Report
Comments and Suggestions for Authors
The similarity index is very high.
Author Response
Thank you for your contribution.
Reviewer 3 Report
Comments and Suggestions for Authors
I appreciate the efforts that authors made to revise their manuscript. However, I feel sorry to say that my previously mentioned points have not been fully addressed. All portions require thorough revision. For example
- Conclusion is citation-based, while it should be derived from your own findings.
- It was not mentioned to add data in the abstract portion; it is suggested to revise it.
- DPN induction is still missing in the material and method section.
- DPN onset was assessed in any group.
- Although the concentration of various PUFAs is modified in response to treatment, it does not support your title.
- The overall quality of language is not satisfactory. As a reference, please have a look on the conclusion of your abstract ( Conclusions: We confirm that omega-3 PUFA are an effective treatment for DPN and sources other than fish oil are beneficial. Ultimate selection of the best source should take into consideration availability, safety, environmental, and economic issues).
- It is Omega-3 PUFAs, when we talk about various types.
- It would suggest presenting your data in any other way instead of just tables.
- It was suggested to use colorful graphs in the previous reports. This point has not been addressed.
Author Response
Response to Reviewer 3:
I appreciate the efforts that authors made to revise their manuscript. However, I feel sorry to say that my previously mentioned points have not been fully addressed. All portions require thorough revision. For example
- Conclusion is citation-based, while it should be derived from your own findings. Response: The conclusion section has been deleted and the revised version addresses only the main findings of the paper. All the editorial comments have been deleted as well as the references in the section.
- It was not mentioned to add data in the abstract portion; it is suggested to revise it. Response: As requested, the abstract has been revised. We now present information relating to the data in the results section.
- DPN induction is still missing in the material and method section. Response: The procedure for inducing diabetes in the rats has been more clearly stated in the Methods section.
- DPN onset was assessed in any group. Response: The onset of diabetes and its assessment is now clearly described in this section.
- Although the concentration of various PUFAs is modified in response to treatment, it does not support your title. Response: The title has been changed and I hope more clearly defines the study.
- The overall quality of language is not satisfactory. As a reference, please have a look on the conclusion of your abstract ( Conclusions: We confirm that omega-3 PUFA are an effective treatment for DPN and sources other than fish oil are beneficial. Ultimate selection of the best source should take into consideration availability, safety, environmental, and economic issues). Response: Additional edits for grammar have been made. The Conclusions section is entirely new.
- It is Omega-3 PUFAs, when we talk about various types. Response: Thank you for bringing this to my attention. You are correct it should be PUFAs. The correction has been made.
- It would suggest presenting your data in any other way instead of just tables. Response: I do not disagree with the reviewer but using tables is the most efficient way to present this extensive amount of data. If I would have provided these data in figures it would have resulted in 16 additional figures. I thought that would be too many and difficult to compare. I respectively request to leave these data in table form except for the vascular data.
- It was suggested to use colorful graphs in the previous reports. This point has not been addressed. Response: I have converted these figures to color as requested.
The changes/edits I have made described above are tracked, but the tracking color remains red for my new edits. I could not determine how to have them highlighted in another color.
Reviewer 4 Report
Comments and Suggestions for Authors
Authors have sufficiently addressed the issues and hence the manuscript may be accepted.
Author Response
Thank you for your contribution.
Round 3
Reviewer 3 Report
Comments and Suggestions for Authors
All issues have been well addressed.